# Botulinum Toxin Injections for Treatment of Drooling in Children with Cerebral Palsy: A Systematic Review and Meta-Analysis

**DOI:** 10.3390/children8121089

**Published:** 2021-11-25

**Authors:** Shang-An Hung, Chung-Lun Liao, Wei-Pin Lin, Jason C. Hsu, Yao-Hong Guo, Yu-Ching Lin

**Affiliations:** 1Department of Physical Medicine and Rehabilitation, National Cheng Kung University Hospital, College of Medicine, National Cheng Kung University, Tainan 704, Taiwan; albert9802064@gmail.com (S.-A.H.); tobyvoids@hotmail.com (W.-P.L.); patchguo@gmail.com (Y.-H.G.); 2School of Medicine, College of Medicine, National Cheng Kung University, Tainan 701, Taiwan; allen01good@gmail.com; 3International Ph.D. Program in Biotech and Healthcare Management, College of Management, Taipei Medical University, Taipei 110, Taiwan; jasonhsu@tmu.edu.tw; 4Department of Physical Medicine and Rehabilitation, College of Medicine, National Cheng Kung University, Tainan 704, Taiwan

**Keywords:** botulinum toxin, drooling, cerebral palsy, sialorrhea, saliva glands

## Abstract

Background: We aimed to review and analyse the effectiveness and safety of botulinum toxin type A (BoNT-A) injections for drooling in children with cerebral palsy. Data sources: We searched the EMBASE, MEDLINE, Cochrane Database of Systematic Reviews, and Cochrane Central Register of Controlled Trials (CENTRAL, The Cochrane Library) databases from inception to January 2020. Methods: We included randomized controlled trials and observational studies which (1) involved children with cerebral palsy, (2) used BoNT-A for control of drooling, and (3) provided quantitative evaluations of drooling before and after intervention with BoNT-A. Results: Twenty-one trials met the inclusion criteria. Most studies showed that BoNT-A injections are safe and efficacious as a treatment for drooling in children with cerebral palsy. Four trials had sufficient data to pool the results for the meta-analysis. Both the drooling quotient (*p* = 0.002) and drooling Ffrequency and severity scale (*p* = 0.004) supported this conclusion. Conclusion: BoNT-A injections are a safe, reversible, effective treatment for drooling control in children with cerebral palsy that can offer effectiveness for more than 3 months with few side effects. The dosage of BoNT-A should not exceed 4 units/kg. Further studies are required to determine the optimal dosage and target glands.

## 1. Introduction

Drooling is the unintentional leakage of saliva from the mouth and is quite common in individuals with cerebral palsy (CP) [1]. Problems associated with drooling may affect people both physically and psychologically. Several approaches to managing drooling have existed for decades, including behavioural interventions, oral and topical medications, irradiation, and surgery [2,3,4,5,6,7]. However, there is currently no satisfactory solution [7]. Physiotherapy and behavioural modification techniques have been used to improve jaw stability, lip closure, and tongue mobility through strengthening of oral motor structures, prompting, and reinforcement [6,7]. However, these methods have unsatisfactory results. The application of benztropine, glycopyrrolate, and scopolamine is limited because of their side effects [3,5]. Furthermore, radiation and surgical procedures are irreversible and may cause permanent adverse events such as xerostomia [4,8].

Botulinum neurotoxin type A (BoNT-A) is a product from the anaerobic bacteria *Clostridium botulinum*. It blocks the effects of acetylcholine at neuromuscular or neuroglandular junctions and inhibits their functions. There are currently three leading botulinum neurotoxin type A products available worldwide: onabotulinumtoxinA (Botox^®^, Allergan Pharmaceuticals, Madison, NJ, USA), abobotulinumtoxinA (Dysport^®^, Biopharm Ltd., Hendy, Swansea, UK), and incobotulinumtoxinA (Xeomin^®^, Merz Pharmaceuticals GmbH, Frankfurt am Main, Germany) [9]. In nature, BoNT-A serotypes are synthesized as macromolecular protein complexes. These protein complexes are referred to as progenitor toxins and consist of nontoxic accessory proteins (NAPs) bonded to the 150 kilodaltons (kD) neurotoxin. The molecular weight of BoNT-A progenitor toxins ranges from 300 to 900 kD depending on the composition of NAPs and the manufacturing process. Botox^®^ and Dysport^®^ are composed of the 150 kD neurotoxin with NAPs, whereas Xeomin^®^ contains only the 150 kD neurotoxin [10]. The conversion ratio between Botox^®^ and Xeomin^®^ is very close to 1:1. In contrast, the conversion ratio between Botox^®^ (or Xeomin^®^) and Dysport^®^ remains controversial, ranging from 1:1 to 1:11 in reported studies [11,12]. However, some studies reported higher efficacy and more adverse events with Dysport^®^ when calculating the dose using a conversion ratio of 1:3, indicating that the true conversion ratio could be lower than 1:3 [13].

Clinical trials have revealed that administration of BoNT-A can effectively and safely reduce excessive muscle tone in children with CP [14,15]. BoNT-A injections to the parotid and/or submandibular glands have been proven to lower the impact of drooling in patients with various neurological disorders, such as CP, [16] Parkinsonism, and others [17]. Nevertheless, the optimal dose and dilution of BoNT-A, the number and type of salivary glands to inject, and the duration of the effect remain unclear, with large variations reported in the literature [17,18].

In this study, we aimed to systematically review and analyse the results of clinical trials investigating the effectiveness and safety of BoNT-A for drooling in individuals with CP.

## 2. Method

### 2.1. Eligibility Criteria

The present review is registered in the PROSPERO database. The search process was performed following the Preferred Reporting Items for Systematic Reviews and Meta-Analyses (PRISMA) guideline. We retrieved studies that met the following criteria (Figure 1): (1) randomised controlled trials (RCTs) or observational studies with more than five participants, (2) full-text articles published in English, (3) inclusion of participants aged 0–18 years with drooling secondary to a definite diagnosis of CP, (4) those on subject injected BoNT-A to the salivary glands as a treatment of drooling, and (5) those with sufficient available quantitative data for a meta-analysis. After screening, 26 full-text articles were assessed for eligibility. We did not enrol two trials due to the administration of botulinum toxin type B as an intervention, [19,20] and three studies were eliminated because they used the same cohort of patients [21,22,23]. Twenty-one articles were enrolled, including five RCTs and sixteen observational studies. The registration number of RCTs we searched for on ClinicalTrials.gov. (https://clinicaltrials.gov, accessed on 1 November 2021) and International Clinical Trials Registry Platform (ICTRP) (https://trialsearch.who.int/AdvSearch.aspx, accessed on 1 November 2021) are listed in Table 1. Five studies were excluded from the meta-analysis because of a lack of raw data from the CP participants, [24,25,26,27,28] and 12 experiments were eliminated because of a lack of data for pooling in the meta-analysis [29,30,31,32,33,34,35,36,37,38,39,40]. Finally, four articles were incorporated into the meta-analysis [16,41,42,43].

### 2.2. Search Strategy

We searched the following databases for publications up to 1 June 2020: EMBASE, Ovid MEDLINE, the Cochrane Database of Systematic Reviews, and Cochrane Central Register of Controlled Trials. Search strategies comprised keywords related to CP, BoNT, children, and drooling.

### 2.3. Study Selection

Two authors (YHG and YCL) retrieved the literature for suitable articles for meta-analysis. After finding and removing duplicates, two investigators (WPL and JCH) checked the eligibility criteria for study inclusion by screening the titles and abstracts of all the studies of interest. Two reviewers (SAH and CLL) assessed the full text of the remaining papers. Discussion among the reviewers was carried out to resolve any discrepancies in the study selection process.

### 2.4. Quality Assessment

The methodological quality of the included studies was evaluated using the Jadad quality score for RCTs and the Newcastle-Ottawa Scale for observational studies. Two authors (SAH and CLL) examined the quality of the included studies, and differences were resolved through consultation with a third reviewer (YCL).

### 2.5. Data Extraction

Extraction of data from the enrolled studies was completed by two of the authors (SAH and CLL). Details of the study design, participant characteristics, the BoNT intervention protocol, and the outcome measures were assessed from each study.

### 2.6. Data Synthesis

Data from RCTs and observational studies were synthesised using Comprehensive Meta-analysis version 2.0 (Biostat Inc., Englewood, NJ, USA). The mean differences with 95% confidence intervals (CIs) were calculated to compare the efficacy between the BoNT-A injection group and the control group. The mean differences and 95% CIs were collected for data before and after interventions with BoNT-A. Because of the significant variance between studies, we used the random effects model. The safety of BoNT-A injection will also be discussed in the text.

## 3. Results

We reviewed a total of 21 articles comprising 827 participants suffering from drooling, including 5 RCTs and 16 observational studies. Among them, 573 (69%) subjects were CP patients with drooling problems. The age of the patients ranged from 1.6 to 21 years. Table 1 lists the characteristics of the participants and the details of the included studies.

Eighteen studies used Botox (onabotulinumtoxinA), and two trials used Dysport (abobotulinumtoxinA) [36,42] for treatment. One article did not mention the BoNT-A brand [26]. The total dose of Botox was in the range 10–120 units or 2–4 units/kg. One study used 100 units of Dysport [42] and in another trial Dysport was injected at 1 units/kg/gland [36]. The injection procedures and types of glands varied among the investigators. Eighteen experiments carried out interventions under ultrasound guidance, and three trials used anatomic landmarks [32,39,42]. BoNT-A was injected into the bilateral submandibular and parotid glands in 13 studies, [24,26,27,29,32,33,34,35,37,38,39,41,43], in six investigations, BoNT-A was injected into the bilateral submandibular glands, [20,23,25,26,31,35] one team injected it into the bilateral parotid glands [42], and one group injected it into a single submandibular gland and the parotid gland on the opposite side [16].

BoNT-A injection was found to offer at least 8–12 weeks of efficacy. One study even showed that the efficacy could last for 6 months [27]. According to our experiences and that of other investigators, repeated injections after 3 months is a better treatment option to avoid recurrence [16,23,24,27,28,32,33,35,40,41,43].

Most studies indicated that BoNT-A injection is a safe, reliable intervention for drooling in patients with CP. Only a few side effects have been reported, including dysphagia, dysarthria, and increased salivary viscosity. However, two studies used a higher dosage (5 units/kg and 8 units/kg) and reported major complications such as aspiration pneumonia, severe dysphagia, and loss of motor control of the head [26,27]. Thus, we suggest that the dosage of BoNT-A injection should not exceed 4 units/kg [26].

Different measurement tools were used between studies, including saliva flow rate [28,31,33,37], bibs per day [30,37,43], drooling impact scale [35,37], drooling rating scale [29], drooling quotient (DQ), [16,23,29,31,34,41,43], drooling frequency and severity scale (DFSS) [16,24,32,35,40,41,42], and so on. In our meta-analysis, DQ and DFSS were selected as the outcome measures, and four trials had usable data that could be pooled into further analyses [16,41,42,43].

The DQ analysis showed a standard difference in means (SMD) of −0.716 (95% CI, −1.17 to −0.263; *p* = 0.002; I^2^ = 80.46; Figure 2). The SMD of the DFSS was −0.888 (95% CI, −1.491 to −0.285; *p* = 0.004; I^2^ = 95.3; Figure 3). Both showed significant improvement in drooling status after intervention with BoNT-A. Only minor transient side effects were reported, including local swelling or pain, increased saliva viscosity, xerostomia, and oral odour; no life-threatening adverse events were noted. These results indicated that BoNT-A injection is an effective, safe treatment in children with CP.

Unfortunately, although we used a random effects model, high heterogeneity was still noted. This may have been a result of variations in the treatment dose, injection site, and an insufficient sample size.

## 4. Discussion

In this paper, using the method of systemic review and meta-analysis, an attempt was made to assess the effectiveness and safety of intervention with BoNT-A in treating drooling in individuals with cerebral palsy. The results showed that BoNT-A injections to the salivary glands are an effective, safe option for management of drooling in children with cerebral palsy. The factors that may influence the results, such as the dosage, the salivary glands selected for the injections, and others, are discussed below.

Objective measures of the amount of drooling and evaluation of its impacts on children have long posed a challenge to clinicians. Multiple questionnaires and scales have been developed for this purpose [44], but none of them are considered the gold standard because saliva production and drooling involve complex mechanisms. The DFSS is the most commonly used measurement tool in the literature and has been shown to be an accurate and quick-to-administer measure of drooling [44]. The DQ is an objective, semiquantitative method for evaluating the severity of drooling that has been proven to be reliable, accurate, and time efficient [45]. Those two methods have also been shown to have high levels of agreement in a recent study [44].

The submandibular, parotid, and sublingual glands are the major salivary glands in humans. They play various roles in physiological conditions [46]. In the fasting state, up to 70% of the saliva is produced by the submandibular glands, 20–25% by the parotid glands, and 5% by the sublingual glands. During meals, saliva is mainly produced by the parotid glands. Submandibular gland secretions are both serous type and mucous type, whereas parotid secretions are mainly serous type [6]. The selection of target salivary glands for treatment of drooling in individuals with cerebral palsy varied between studies, with most researchers injecting the parotid and/or the submandibular glands on both sides. There is currently no clinical guideline available on the choice of glands for the administration of BoNT. Suskind and Tilton [29] reported that injection of both parotid and submandibular glands offers better results than targeting the submandibular glands only, whereas recent studies showed that injecting only the submandibular glands could offer an optimal effect under ultrasound treatment [47,48]. One team chose the unilateral submandibular gland and the parotid gland of the opposite side in the hope of keeping at least 50% of the fasting and postprandial saliva production [16]. The authors hypothesised that such a management strategy may induce a more balanced decrease in the production of serous type saliva and mucous type saliva. Dry mouth or other adverse events after intervention with BoNT-A could possibly be prevented by avoiding a sharp decrease in the volume of either type of saliva [16]. To determine whether such a protocol actually decreases the possibility of xerostomia and side effects related to ablation of saliva, further studies are needed.

Some studies assessed the differences between BoNT interventions and other management methods used to control the problem of drooling. One clinical trial compared BoNT injections with scopolamine [28] and showed no significant outcome differences between the two treatments. However, moderate to severe side effects were detected in 71.1% of the patients who received treatment with scopolamine. BoNT injections demonstrated only non-severe side effects. Two studies investigated the differences in influences between surgical intervention and BoNT injections. One researcher reported that bilateral submandibular duct ligation is more effective than BoNT injection, which may cause more complications and morbidities [40]. Another study found that submandibular duct relocation offered greater and longer-lasting results [49]. However, the authors mentioned that performing surgery could be premature because salivary control in children with developmental disorders may spontaneously improve. In this situation, BoNT injections could provide advantages over surgery because administration of BoNT for drooling is not a permanent procedure and is reversible if there are complications [23].

Only three teams used anatomic landmarks as the intervention technique. Although a systematic review reported that no advantage was conferred in injecting the glands under ultrasound guidance [50], the accuracy of the targeted glands will affect the efficacy theoretically. Thus, ultrasound guidance when performing injections is preferable in future studies.

The optimal dosage of BoNT-A injections for managing drooling remains a subject of controversy. Larger doses of BoNT-A led to higher major complication rates. For example, Chan et al. adopted 8 units/kg [26] and Khan et al. used 5 units/kg [27] in their studies. They found 4.2% and 5.5% adverse events, respectively. Chan et al. [26] suggested that to ensure safety, the BoNT-A dosage should not be more than 4 units/kg. One study reported that repeated BoNT injections may cause gland atrophy and a decrease in functionality [51]. However, in another trial BoNT was injected six times, and dose and gland involvement were shown to increase gradually, with no obvious functional decline or severe side effects noted [37].

BoNT-A and BoNT-B showed similar efficacy in management of drooling in patients with Parkinson’s disease or amyotrophic lateral sclerosis [52,53]. However, there are two reasons that we did not include BoNT-B in our review. First, the conversion ratio between BoNT-A and BoNT-B is still debated. It ranges from 1:30 to 1:75 [54,55,56], making it difficult to pool data into meta-analysis. Second, although one study encouraged the use of BoNT-B for management of drooling due to the higher rates of dry mouth compared with BoNT-A [57], some research reported higher risks of autonomic side effects for BoNT-B injection such as dysphagia, thickness of saliva, and constipation [58]. Higher immunogenicity with BoNT-B was also reported in one study [59,60]. Taking the above factors into consideration, we believe that BoNT-A injection is more preferable to BoNT-B for children with cerebral palsy and sialorrhea.

This study has several limitations. First, as is the case with most meta-analyses, there is a potential for publication bias. Studies that fail to show the benefits of BoNT-A injections may not be reported or remain unpublished and thus cannot be searched for in the literature and will not have been included in this meta-analysis. Second, we reviewed only articles published in English; therefore, some experiments in other languages might have been ignored, which could possibly have influenced the results. Third, many RCTs did not meet the inclusion criteria and were thus excluded from this study, making the number of studies available for analysis relatively few. Fourth, high heterogeneity was noted due to variations in treatment dosages and methods. Finally, small patient numbers in most studies and variable outcome measurements in the English literature made it difficult to conduct a comprehensive meta-analysis.

## 5. Conclusions

BoNT-A injection is a safe, reversible, effective treatment for drooling in children with CP, with few adverse effects and no life-threatening events. The duration of effectiveness could be a minimum of 8–12 weeks, and we suggest repeating injections after 3 months. The dosage of BoNT-A should not exceed 4 units/kg. Further studies are needed to determine the optimal dosage and target glands.

## Figures and Tables

**Figure 1 children-08-01089-f001:**
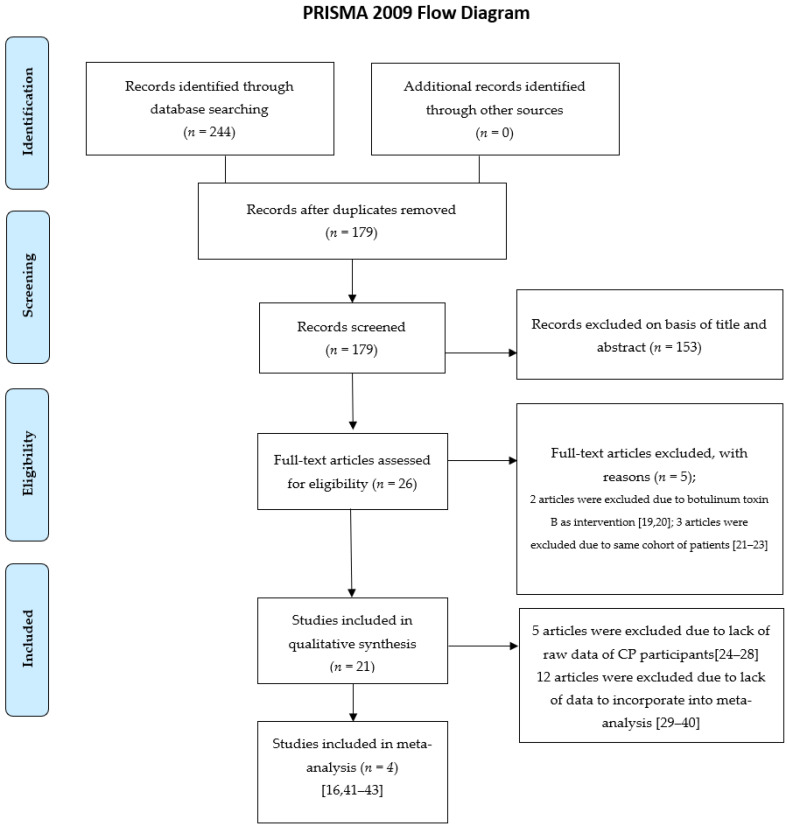
Included and excluded studies.

**Figure 2 children-08-01089-f002:**
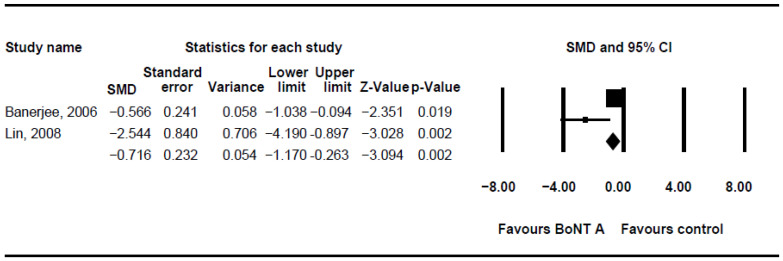
Meta-analysis of drooling quotient of botulinum toxin to salivary gland for the treatment of drooling in children with cerebral palsy. SMD, standard difference in means; CI, confidence interval.

**Figure 3 children-08-01089-f003:**
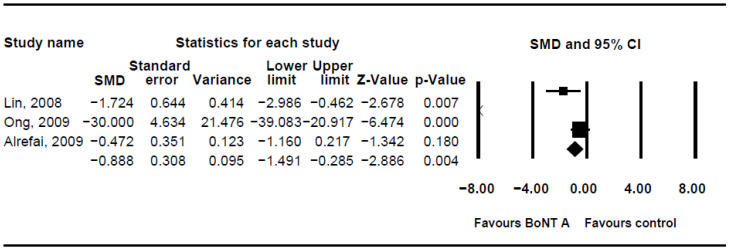
Meta-analysis of drooling frequency and severity scale of botulinum toxin to salivary gland for the treatment of drooling in children with cerebral palsy.

**Table 1 children-08-01089-t001:** Population characteristics of included studies.

Study	Treatment (n)	Mean Age, y (range)	Diagnosis	Injection Type and Dose	Ultrasound Guidance	Site	Registration Number of RCT
**Randomized control trials**							
Lin et al., 2008 [16]	13	14.2	CP	Botox, 2 U/kg/gland	Y	One PG and the contralateral SMG gland	NCT00173745
Alrefai et al., 2009 [42]	24	3.5 (1.75–7)	CP	Dysport, 100 U	N	Bil. PG	Not provided
Wu et al., 2011 [33]	20	3–16	CP	Botox, 30–50 U	Y	Bil. PG and SMG	Not provided
Nordgarden et al., 2012 [34]	6	13.7 (10–18)	CP	Botox, 25 U/gland	Y	Bil. PG and SMG	Not provided
Bekkers et al., 2019 [40]	26	11	CP DD	Botox 25 U/gland	Y	Bil. SMG	NTR3537
**Observational studies**							
Suskind et al., 2002 [29]	22	8–21	CP	Botox, 10–70 U	Y	Bil. PG and SMG	
Jongerius et al., 2004 [28]	45	9.5 (3–16)	CP	Botox, 30–50 U	Y	Bil. SMG	
Savarese et al., 2004 [30]	21	5–18	CP	Botox, 15 U/gland	N	Bil. SMG	
Banerjee et al., 2006 [41]	19	10.8 (6–16)	CP	Botox, 2 U/kg, max 70 U	Y	Bil. PG and SMG	
Ong et al., 2009 [43]	21	8.4 (4–12)	CP	Botox, 60–80 U	Y	Bil. PG and SMG	
Scheffer et al., 2010 [23]	131	10.9	CP PR	Botox, 30–50 U	Y	Bil. SMG	
Khan et al., 2010 [27]	45	10.5	CP Other	Botox, max 5 U/kg	Y	Bil. PG and SMG	
Erasmus et al., 2011 [31]	126	10 y 11 m	CP ID	Botox, 50 U	Y	Bil. SMG	
Nicola et al., 2011 [32]	9	9.3 (5–17)	CP OS	Botox, 30–50 U	N	Bil. PG and SMG	
Tiigimäe-Saar et al., 2012 [35]	12	1.6–11	CP	Botox, 2 U/kg	Y	Bil. PG and SMG	
Sidebottom et al., 2013 [36]	30	11 (4–17)	CP NS	Dysport, 1 U/kg/gland	Y	Bil. SMG	
Chan et al., 2013 [26]	69	9.5	CP Other	BoNT A ^a^, 8 U/kg	Y	Bil. PG and SMG	
Møller et al., 2015 [37]	14	9	CP	Botox, 20–120 U	Y	Bil. PG and SMG	
Matthew et al., 2016 [38]	111	7	CP NS	Botox, 100 U	Y	Bil. PG and SMG	
Sürmelioğlu et al., 2018 [39]	27	11.5 (6–16)	CP	Botox, 60 U	N	Bil. PG and SMG	
Gubbay et al., 2019 [24]	15	9.9 (3–14)	CP Other	Botox, 1 unit/kg/gland, max: 100 U	Y	Bil. PG and SMG	

^a^ Not mentioned the product name; Bil., bilateral; BoNT A, botulinum toxin type A; CP, cerebral palsy; DD, developmental disability; ID, intellectual disability; NS, not specified; OS: Opercular Syndrome; PG, parotid gland; PR, psychomotor retardation; RCT, randomized control trial; SMG, submandibular gland.

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
