# Peer review of "Botulinum Toxin Injections for Treatment of Drooling in Children with Cerebral Palsy: A Systematic Review and Meta-Analysis"

_children, 2021, doi:10.3390/children8121089_

Round 1

Reviewer 1 Report

Dear Editor,

thank you for the opportunity to review this interesting manuscript about the use of botulinum toxin in children with cerebral palsy with sialorrhea. 

The topic is relevant because it gives us a comprehensive review of botulinum toxin use in sialorrhea patients with cerebral palsy. The papar is well written and easy to read. The conclusions are consistent with the presented evidence and arguments and address to the posed main question.

Best regards.  

Author Response

Thank you very much for your review.

Reviewer 2 Report

congratulations to the authors for the well-written article. Please answer the following comments.

  1. There are only 4 articles eligible for the meta-analysis, which is a very limited number of articles to come to a conclusion about the efficiency of effectiveness, and safety of botulinum 19n toxin type A (BoNT-A) injection in cerebral palsy.
  2. can you please explain in detail the 3 types of BoNT-A?
  3. Does onabotulinumtoxinA FDA approve for treating sialorrhea in children with Cerebral palsy? is it an off-label use?
  4. Please comment about rimabotulinumtoxinB use in sialorrhea in CP children and is it FDA approved? Also please cite any article if available.
  5. The reference article 21 stated major complications in 5 patients who were hospitalized after BoNT-A injection. Can the authors explain how they came to a conclusion the use of BoNT-A is safe with very few side effects?

Reviewer 3 Report

I consider the article to be of great importance, it has been easy for me to read and it is well structured. The conclusions are related to the objectives and highlight the importance of the topic and the need to continue researching it to improve the scientific evidence. I have no extra considerations for authors 

Author Response

Thank you very much for your review

Round 2

Reviewer 2 Report

Thank you for the review